Within-host adaptive speciation of commensal yoyo clams leads to ecological exclusion, not co-existence

Harrison Teal A. tealh@umich.edu 1
Goto Ryutaro 2
Li Jingchun 3
Ó Foighil Diarmaid diarmaid@umich.edu 1
1 Department of Ecology and Evolutionary Biology and Museum of Zoology, University of Michigan , Ann Arbor , MI , United States of America
2 Seto Marine Biological Laboratory, Field Science Education and Research Center, Kyoto University , Shirahama , Wakayama , Japan
3 Department of Ecology and Evolutionary Biology and Museum of Natural History, University of Colorado , Boulder , CO , United States of America
Garant Dany
Electronic publication date: 2024 Aug 5
Publication date: 2024
Volume: 12
Electronic Location ID: e17753
Received 2024 Mar 4; Accepted 2024 Jun 25
Copyright: ©2024 Harrison et al.
Copyright year: 2024
Copyright holder: Harrison et al.
License: This is an open access article distributed under the terms of the Creative Commons Attribution License, which permits unrestricted use, distribution, reproduction and adaptation in any medium and for any purpose provided that it is properly attributed. For attribution, the original author(s), title, publication source (PeerJ) and either DOI or URL of the article must be cited.
License URL: https://creativecommons.org/licenses/by/4.0/

Keywords: Commensalism, Competitive exclusion principle, Neutral theory, Within-host speciation, Galeommatoidea, Lysiosquilla scabricauda, Divariscintilla, Indian River Lagoon, Yoyo clams, Mantis shrimp

Funding: University of Michigan EEB Block Grant Smithsonian Minority Awards Fellowship Program American Microscopical Society Microscopy Training Fellowship University of Michigan Rackham Graduate School Research Grant University of Michigan Museum of Zoology John B. Burch Malacology Fund This study is based on Teal A. Harrison’s Masters’ Thesis and was supported by a University of Michigan EEB Block Grant, the Smithsonian Minority Awards Fellowship Program, the American Microscopical Society Microscopy Training Fellowship, a University of Michigan Rackham Graduate School Research Grant, and by the University of Michigan Museum of Zoology John B. Burch Malacology Fund. There was no additional external funding received for this study. The funders had no role in study design, data collection and analysis, decision to publish, or preparation of the manuscript.

==============================
Symbionts dominate planetary diversity and three primary symbiont diversification processes have been proposed: co-speciation with hosts, speciation by host-switching, and within-host speciation. The last mechanism is prevalent among members of an extraordinary marine symbiosis in the Indian River Lagoon, Florida, composed of a host mantis shrimp, Lysiosquilla scabricauda, and seven host-specific commensal vasconielline “yoyo” clams (Galeommatoidea) that collectively occupy two distinct niches: burrow-wall-attached, and host-attached/ectocommensal. This within-host symbiont radiation provides a natural experiment to test how symbiont coexistence patterns are regulated in a common ancestral habitat. The competitive exclusion principle predicts that sister taxa produced by adaptive speciation (with distinct morphologies and within-burrow niches) are most likely to coexist whereas the neutral theory predicts no difference among adaptive and non-adaptive sister taxa co-occurrence. To test these predictions, we engaged in (1) field-censusing commensal species assemblages; (2) trophic niche analyses; (3) laboratory behavioral observations. Although predicted by both models, the field census found no mixed-niche commensal assemblages: multi-species burrows were exclusively composed of burrow-wall commensals. Their co-occurrence matched random assembly process expectations, but presence of the single ectocommensal species had a highly significant negative effect on recruitment of all burrow-wall commensal species (P < 0.001), including on its burrow-wall commensal sister species (P < 0.001). Our stable isotope data indicated that commensals are suspension feeders and that co-occurring burrow-wall commensals may exhibit trophic niche differentiation. The artificial burrow behavioral experiment yielded no evidence of spatial segregation among burrow-wall commensals, and it was terminated by a sudden breakdown of the host-commensal relationship resulting in a mass mortality of all commensals unattached to the host. This study system appears to contain two distinct, superimposed patterns of commensal distribution: (1) all burrow-wall commensal species; (2) the ectocommensal species. Burrow-wall commensals (the plesiomorphic condition) broadly adhere to neutral theory expectations of species assembly but the adaptive evolution of ectocommensalism has apparently led to ecological exclusion rather than coexistence, an inverse outcome of theoretical expectations. The ecological factors regulating the observed burrow-wall/ectocommensal exclusion are currently obscure but potentially include differential recruitment to host burrows and/or differential survival in “mixed” burrow assemblages, the latter potentially due to changes in host predatory behavior. Resampling host burrows during commensal recruitment peak periods and tracking burrow-wall commensal survival in host burrows with and without added ectocommensals could resolve this outstanding issue.

Introduction

A striking feature of life on Earth is its high degree of ecological nestedness, a condition famously satirized by Swift (1733): “So, naturalists observe, a flea hath smaller fleas that on him prey, and these have smaller still to bite ’em; and so proceed ad infinitum”. An important but often-overlooked consequence of this feature is that most species are in fact symbionts (parasites/commensals/mutualists) whose habitats consist of other (host) species (Windsor, 1998; Poulin & Morand, 2004; Moran, 2006).

Symbiont diversification processes have therefore played outsized roles in generating our planet’s fundamental biodiversity and two main generative evolutionary mechanisms have been proposed: co-speciation with hosts, and speciation by host-switching (e.g., Ricklefs, Fallon & Bermingham, 2004). The former mechanism is host-driven—host lineage speciation events lock-stepping symbiont lineage speciation—and it is thought to be highly prevalent in nature, e.g., co-speciation of bacterial endosymbionts with insect hosts alone may form the bulk of all speciation events (Larsen et al., 2017; Hernández-Hernández et al., 2021). The latter mechanism is symbiont-driven—colonization of new hosts providing new ecological portals for symbiont speciation—and it has been proposed as a major driver of speciation in both terrestrial (Coyne & Orr, 2004; Matsubayashi, Ohshima & Nosil, 2010) and marine (Duffy, 1996; Goto et al., 2012; Hurt et al., 2013; Fritts-Penniman et al., 2020; Rodriguez & Krug, 2022) biotas.

A third diversification mechanism, within-host speciation, has received less attention and it involves the evolution of sister species that retain the same ancestral host. Co-existence of sister taxa on their host might a priori be expected to approximate neutral theory (Hubbel, 2001) expectations of species assembly because of their joint persistence within a shared, highly specialized ancestral habitat. However, competitive exclusion principle-based perspectives (Grinnell, 1904; Hardin, 1960; Chesson, 2003) have dominated species diversity studies over the past century (Simha, Hoz & Carley, 2022). This is also apparent for within-host speciation case histories where niche differentiation or allopatry is implicitly expected, e.g., within-host phytophagus insect speciation studies emphasize cases of either adaptive speciation, e.g., specialization for discrete host tissues (Cook et al., 2002; Joy & Crespi, 2007; Althoff, 2014), or discrete host life history stages (Zhang et al., 2015), or non-adaptive allopatric speciation occurring in exclusive subsets of a host range (Imada, Kawakita & Kato, 2011). In contrast, there have been few studies of sympatric, ecologically non-differentiated sister species that share the same host.

Almost all evolutionary radiations have the potential to produce new members through either adaptive or non-adaptive speciation processes (Czekanski-Moir & Rundell, 2019; Matsubayashi & Yamaguchi, 2022). In principle, a within-host symbiont radiation that contained sympatric sister species pairs respectively generated by adaptive and by non-adaptive speciation processes could represent an ideal natural experiment to test how symbiont coexistence patterns are regulated in a common ancestral habitat. The competitive exclusion principle (Grinnell, 1904; Hardin, 1960; Chesson, 2003) predicts that sister taxa produced by adaptive speciation and occupying distinct host niches are most likely to coexist on individual hosts. In contrast, the neutral theory (Hubbel, 2001) predicts that sister taxa produced by non-adaptive speciation and having similar host niches are equally likely to coexist with each other as with their ecologically differentiated co-symbionts.

A single-host marine symbiont assemblage documented in the Indian River Lagoon (IRL) on the eastern coast of Florida exhibits many of the model attributes outlined above. The host, Lysiosquilla scabricauda (Lamarck, 1818), is a benthic ambush predator “spearing” mantis shrimp (Caldwell & Dingle, 1976; DeVries, Murphy & Patek, 2012) that lives in large burrows up to 10 m in length within sandy substrates (Christy & Salmon, 1991) and is widely distributed in the western Atlantic from the southeastern USA to southern Brazil (Tavares, 2002; Reaka et al., 2009). In the IRL, L. scabricauda hosts seven species of commensal galeommatoidean bivalves currently placed in two vasconielline genera—Divariscintilla (six species) and “Parabornia” (one species)—that appear to be host-specific, i.e., are known to occur only within L. scabricauda burrows (Mikkelsen & Bieler, 1989; Mikkelsen & Bieler, 1992; Goto, Harrison & Ó Foighil, 2018). The six species of Divariscintilla (D. yoyo Mikkelsen & Bieler, 1989, D. troglodytes Mikkelsen & Bieler, 1989, D. octotentaculata Mikkelsen & Bieler, 1992, D. luteocrinita Mikkelsen & Bieler, 1992, D. cordiformis Mikkelsen & Bieler, 1992, and a new, undescribed species D. aff. yoyo (Goto, Harrison & Ó Foighil, 2018)) are currently known only from the IRL and nearby Floridian locations (Mikkelsen & Bieler, 1992; Mikkelsen, Mikkelsen & Karlen, 1995). All six attach to the smooth, hard-packed host burrow walls (Fig. 1) via a long, thin posterior foot extension that secretes anchoring byssus threads, and contraction/relaxation of this “hanging foot” structure produces characteristic yoyo-like movements (Mikkelsen & Bieler, 1989; Mikkelsen & Bieler, 1992; Goto, Harrison & Ó Foighil, 2018; Movie S1)—hence the informal “yoyo clams” moniker (Mikkelsen & Bieler, 1989). In contrast, the “Parabonia” species, “P”. squillina Boss, 1965, is an ectocommensal, attaching directly to the host (Fig. 1), specifically to the lateral portion of its pleonal sternite (Goto, Harrison & Ó Foighil, 2018). Its known range extends from Panama to Florida (Boss, 1965; Moore & Boss, 1966; Abbott, 1974) and it has one very similar ectocommensal congener, “P”. palliopillata Simone, 2001, recorded from southern Brazilian L. scabricauda host populations (Simone, 2001; Goto, Harrison & Ó Foighil, 2018).

Figure 1 Schematic section of a composite Indian River Lagoon Lysiosquilla scabricauda burrow.

This shows the relative positioning (by yabby pump field sampling) of the five burrow-wall commensal species (Divariscintilla spp.), and the single ectocommensal (“Parabornia” squillina) species, collected in this study. Also shown, in outline, are the inferred phylogenetic relationships of the 6 IRL commensals (Goto, Harrison & Ó Foighil, 2018). This is an original artwork by John Megahan apart from the commensal clam thumbnail photographs that were sourced from Goto, Harrison & Ó Foighil (2018) with permission from Oxford University Press.

A vasconielline molecular phylogenetic analysis (Goto, Harrison & Ó Foighil, 2018) illuminated the evolutionary relationships among 6/7 of the IRL L. scabricauda commensals (the rarest species, D. cordiformis, was unavailable for genotyping). One species, D. troglodytes, was phylogenetically distinct and placed topologically among Pacific Ocean burrow-wall lysiosqillid commensals, implying that its presence in L. scabricauda burrows involved an ancestral host-switching event coupled with inter-ocean basin migration. The remaining five L. scabricauda commensals formed a host-specific clade, a result consistent with within-host speciation, but not necessarily in sympatry as initial differentiation may have occurred in allopatry (Rundell & Price, 2009), i.e., in discrete subsets of the host’s extensive western Atlantic range (Goto, Harrison & Ó Foighil, 2018). The host-specific clade contained two well-supported clade tip sister relationships. One involved a cryptic sister species pair of burrow-wall commensals—D. yoyo and D. aff. yoyo—that are apparent products of non-adaptive speciation. They are effectively identical in external appearance and in within-burrow habitat but can be distinguished morphologically by details of their (mantle-covered) anterior shell margins, in addition to their gene sequences (Goto, Harrison & Ó Foighil, 2018). The other comprised D. octotentaculata, a burrow-wall commensal, and “P”. squillina, the ectocommensal, two species that differ not only in within-burrow habitat but also in many aspects of their morphologies. Goto, Harrison & Ó Foighil (2018) concluded that “P”. squillina was a product of adaptive speciation and ecological character displacement (Grant & Grant, 2006) from a burrow-wall commensal common ancestor with D. octotentaculata. This evolutionary process involved an ecological shift to an ectocommensal niche along with a suite of associated morphological changes: loss of specialized “hanging foot” structures, loss of hypertrophied mantle tissue enveloping the shell, loss of prominent sensory tentacles, as well as gain of specialized mantle margin papillae. Note that the phylogenetic placement of “P”. squillina firmly within the Divariscintilla clade (Goto, Harrison & Ó Foighil, 2018) calls into question its current generic designation and we therefore place that designation in quotations pending a formal generic revision.

Mikkelsen & Bieler’s (1992) focus was primarily taxonomic, but they also commented on the relative frequency of burrow-wall commensals recovered from individual IRL L. scabricauda burrows. Most burrows with these commensals contained D. octotentaculata, usually in combination with one or more of four congeners: D. yoyo, D. troglodytes, D. luteocrinita, and D. cordiformis. They concluded that “no ecological niche separation between the five sympatric species was recognized, leaving interesting questions for future research”. Mikkelsen & Bieler (1992) did not provide data on the frequency of the ectocommensal “P”. squillina in IRL host burrows but anecdotally noted that it “has not been collected in burrows containing Divariscintilla species”.

These preliminary ecological observations (Mikkelsen & Bieler, 1992) are broadly consistent with neutral theory (Hubbel, 2001) expectations for IRL commensal vasconielline species. Our aim in this study was to revisit this issue in light of the new evolutionary relationships data among the commensals, including the sister species pairs produced by adaptive (D. octotentaculata and “P”. squillina) and non-adaptive (D. yoyo and D. aff. yoyo) within-host speciation (Goto, Harrison & Ó Foighil, 2018). We employed a diversity of approaches including (1) censusing commensal species assemblages in host burrows; (2) testing for dietary differentiation via isotope composition analyses; (3) laboratory behavioral observations of artificial burrow commensal assemblages with, and without, hosts. Although our results are the inverse of competitive exclusion expectations, they are also not fully consistent with neutral theory predictions, and they imply that the adaptive evolution of ectocommensalism may have disrupted ancestral coexistence modalities among members of this host-specific commensal community.

Methods

Sampling sites

From June 14th to July 25th, 2017, the first author (after obtaining a Florida state collecting permit) performed low tide field sampling of 86 L. scabricauda burrows at five adjacent shallow water sandflat study sites within the IRL’s Ft. Pierce Inlet (Fig. 2). The three sites (1, 4 & 5) on the northern margin of the Inlet (Fig. 2) collectively contain the type localities of five commensal clam species: Divariscintilla octotentaculata, D. luteocrinita, D. troglodytes, D. yoyo, and D. cordiformis (Mikkelsen & Bieler, 1989; Mikkelsen & Bieler, 1992).

Figure 2 Maps of the field study sites.

The small inset map (top left) shows the position of the Indian River Lagoon (IRL) study area on the eastern coast of Florida. The main map illustrates the five intertidal study field sites flanking the IRL’s Fort Pierce Inlet. Both are based on Google Maps images annotated by John Megahan in compliance with fair use guidelines.

Host burrow identification and host capture

During each field session, visible host burrows covered by ∼5–10 cm of seawater within the targeted study site were flagged for sampling. These typically represented <20% of the Lysiosquilla scabricauda burrows (characterized by their irregularly square openings, about one cm2in area, often covered by a sand cap that was differentially textured from the surrounding sand flat) visible within each intertidal site. L. scabricauda specimens were collected manually using a bait-and-capture technique (Goto, Harrison & Ó Foighil, 2018; Fig. 3). A bait fish was placed directly over a submerged burrow opening and held for a 3-minute trial period to elicit an attack by a resident stomatopod. If no host response occurred during that interval—approximately 50% of the burrows investigated—a new host burrow was then attempted. The raptorial appendages of lysiosquillid stomatopods are barbed, razor sharp, and designed to impale soft-bodied prey (DeVries, Murphy & Patek, 2012). Thus, thick fishing gloves were worn for protection and once a resident stomatopod impaled the bait or otherwise presented at the mouth of the burrow, its raptorial appendages were sequentially grasped by hand (Fig. 3) and held firmly as it attempted to pull itself downward into its burrow. As the restrained stomatopod tired, it was slowly pulled upward out of the burrow. A large majority of collected host specimens lacked ectocommensal clams and these were released within 5 min of capture.

Figure 3 Field photograph showing capture of a host Lysiosquilla scabricauda specimen.

The first author is shown firmly grasping the host specimen’s two raptorial appendages prior to carefully lifting it out of its flagged burrow opening. A video recording of an entire host capture sequence is available at figshare: Ó Foighil (2024). Mantis Shrimp Capture Technique.m4v. figshare. Media. https://doi.org/10.6084/m9.figshare.24847938.v1.

Collecting burrow-wall commensal clams

Once a host L. scabricauda had been collected, its burrow was then sampled for yoyo clam burrow wall commensals using a stainless-steel bait pump (“yabby pump”). As emphasized by Mikkelsen & Bieler (1989) and Mikkelsen & Bieler (1992), this method effectively samples only its own length (0.5–1.0 m) of the vertical parts of the stomatopod’s U-shaped burrow, leaving the deeper horizontal section unsampled. The contents of single pulls of the yabby pump were expelled into a 2 mm sieve and this process was repeated until three pulls failed to return any observable clams, or until the vertical arm of the host burrow collapsed from the repeated suctioning. Regardless of species, yoyo clams were readily recognized by their characteristic off-white, mucoid appearance against the mesh and the residual sediment particles retained in the sieve. Individual clams were carefully picked up using a feather weight forceps and placed into 50 ml tubes of seawater. Any ectocommensal clams detected on the stomatopod host were similarly detached from the base of the host pleopods and placed in seawater tubes. Back in the laboratory, all live commensal clams sampled from individual host burrows were maintained together in burrow-specific, labelled finger bowls containing filtered sea water with slight aeration. All clams were then identified to species using a dissecting microscope and their mantle lengths measured. Species counts and the number of individuals per species were recorded for each burrow sample.

Statistical analyses of co-occurrence data

Several statistical tests were performed on the commensal species frequency data. These included over-dispersion tests for the two most frequent species (“P”. squillina and D. octotentaculata) using the “overdispersion.test” function in R 4.3.1, to determine if the observed individual distributions in host burrows were more clustered than expected by chance.

In addition, we conducted several simulations tests of the co-occurrence patterns of the different commensal species to determine if they fall within the expectation of a random larval settling process. IRL Divariscintilla species have “mixed” larval development in which early developmental stages are ctenidially-brooded, then released into the water column as early, straight-hinged “D” veligers to undergo an obligate period of planktotrophic larval development and dispersal, thereby greatly reducing the likelihood of resettlement in parental burrows (Mikkelsen & Bieler, 1989; Mikkelsen & Bieler, 1992). The details of “Parabornia” squillina’s early development are currently unknown but its prodissoconch structure is consistent with it also having an obligate planktototrophic larval dispersal phase (Fig. S1). (Similarly, presence of large numbers of small (<100 µm) brooded “ova” in the ctenidia of its Brazilian congener, “P”. palliopapillata, (Simone, 2001; Fig. 13 therein) is indicative of planktotrophic larval development in Galeommatoidea (Ó Foighil, 1988)).

We therefore assumed that each IRL commensal clam was independently recruited to its host burrow from a planktonic pool of metamorphosing veliger larvae. We were particularly interested in testing for settlement/survival effects among the ectocommensal “P”. squillina and the burrow-wall commensal Divariscintilla species. The competitive exclusion principle (Grinnell, 1904; Hardin, 1960; Chesson, 2003) predicts that IRL Divariscintilla species (which all share the same burrow-wall niche) will co-occupy burrows host less frequently with each other that with the niche-differentiated ectocommensal “P”. squillina. In contrast, the neutral theory of species assembly (Hubbel, 2001) predicts that each member of the IRL commensal vasconielline community will occupy host burrows irrespective of the presence or absence of any other member, and that the observed co-occurrence pattern will therefore match random larval settling expectations.

Expected co-occurrence distributions of different species pairs were generated by randomly allocating clams to burrows based on the observed proportions of each species across all burrows. For example, there was a total of 73 D. octotentaculata and 20 “P”. squillina in all burrows, meaning 78% (73/(73 + 20)) were D. octotentaculata and 22% were “P”. squillina. The two species were then randomly allocated to all burrows based on this probability, keeping the total clam count in each burrow equal to the observed value (i.e., if a burrow had five clam individuals, then simulation was done five times for that burrow). This process was repeated 1,000 times. After the simulation, numbers of burrows where two species co-occurred were counted and summarized in a histogram. The actual observed number of co-occurred burrows was also plotted on the histogram to compare with the theoretical distribution. P-values were calculated by the percentile of the actual observed value in the simulated distribution. This comparison was performed between “P”. squillina and all wall-commensal species (treated as the same type); “P”. squillina and D. octotentaculata; “P”. squillina and D. luteocrinita; and D. octotentaculata and D. leuteocrinita. The other wall commensal species had low occurrences therefore they were not compared with “P”. squillina individually.

Stable isotope analyses

An organism’s stable isotope composition is shaped by, and indicative of, its diet/trophic niche (Layman et al., 2007). To test if the IRL L. scabricauda vasconielline commensal species differ in their trophic niches, 18 burrow-wall commensals (11 D. octotentaculata, 4 D. luteocrinita, 2 D. yoyo and 1 D. troglodytes) and 20 “P”. squillina ectocommensals, together with samples of within-burrow potential basal trophic resources—tissue from 19 L. scabricauda specimens, suspended particulate organic matter from 35 burrows, and deposited organic matter from 34 burrows—were collected (Fig. 4) to measure their respective isotopic niche widths.

Figure 4 Sampling scheme for stable isotope analyses.

Schematic diagram of a composite Indian River Lagoon mantis shrimp Lysiosquilla scabricauda host burrow showing the four primary burrow components sampled for isotope analyses: individual commensal clams, comprising both burrow-wall and ectocommensal species (3), and their potential basal trophic resources (deposited organic matter (1), suspended particulate organic matter (2), and host tissue (4)). This figure is an original artwork by John Megahan.

The clams were housed separately in petri dishes of filtered sea water for 12 h to allow for their gut contents to empty, after which their soft tissues were separated from their shells prior to further processing. Host stomatopod specimens were euthanized in an ice-water slurry. To sample burrow water particulate organic matter (POM) the flow of ambient water into burrows was first blocked with a cylindrical barrier (bucket with the bottom removed) enclosing the burrow opening. For each burrow, 1 liter of burrow water was field-collected with a large syringe (with a one mm filter attachment) and filtered in the laboratory onto 4.7 cm diameter Whatman GF/F glass microfiber filters using a six-manifold filtration system. Deposited organic matter was sampled by collecting the oxygenated layer of burrow wall sediment with a shallow teaspoon. All commensal tissue specimens and potential basal resource samples were lyophilized using a Labconco Freeze Dry System prior to further processing. This involved grinding the commensal samples, and the POM samples (first removed from the filter paper with a spatula), with disposable mortar and pestles, grinding the right merus of each host specimen for 4 min in a ball-mill grinder, and grinding each sediment sample for 4 min in a bead-mill grinder. Approximately 2.5 mg of each ground host and commensal species sample was weighed into individual 5 × nine mm pressed tin capsules. Approximately 5–7 mg of each POM and 35–40 mg of each sediment sample were weighed in 10.5 × 9 pressed, light-weight silver capsules and acidified with reagent grade HCL to remove any undetected shell fragments. All samples were analyzed at the Center for Applied Isotope Studies at the University of Georgia for carbon and nitrogen isotopic signatures. The isotopic niche width of each species was quantified as Standard Ellipse Areas (SEAB) which estimate mean population-level isotopic niche spaces while accounting for variation in population size, in the R package SIAR (Jackson et al., 2011).

Artificial burrow construction & observations

Two observable, artificial stomatopod burrows were constructed at the Smithsonian Marine Station at Fort Pierce. Each was positioned within a 110 L flow-through glass aquarium tank that was partitioned with a sheet of PVC to form a nine cm wide cross-section of sediment observable through the front facing wall of the tank. The PVC barrier was 10 cm shorter than the tank, allowing flow to pass across it, and it was kept in place by a series of 3.8 cm diameter PVC tubes (Fig. S2). To construct artificial stomatopod burrows, a 3.8 cm diameter electrical conduit tube was first planed in half and then taped to form a U shape. The inner surfaces of the tube were lightly coated with silicone rubber sealant, packed with dry sand, and allowed to set overnight. The unattached sand was then removed, and the sand-coated tube halves were individually placed into separate aquarium tanks with the cut edge held in place against the tank walls with additional dry sand and the tube openings flush with the sand surface (Fig. 5). The aquaria were then filled with aerated sea water and allowed to settle for 24 h.

Figure 5 Aquarium artificial host burrow for viewing commensal/host interactions in the laboratory.

Note the host Lysiosquilla scabricauda (arrow) within the sand-coated PVC artificial burrow structure. This photo was taken prior to the addition of commensals.

A host Lysiosquilla scabricauda was introduced to one of the aquaria, was acclimated for a week, and offered shrimp and small fish as food. It entered the artificial burrow on the first day and remained in the burrow for the duration of the experiment (Fig. 5). During the acclimation period, the stomatopod used loose sand in the tank to form a cap for the artificial burrow and consistently maintained this cap by collecting excess sand in the burrow with its maxillipeds. The other aquarium was kept host-free.

An experimental community of four species (D. octotentaculata, D. luteocrinita, “P”. squillina, and D. troglodytes) were introduced, in proportions approximate to their natural IRL frequencies, to both tanks. The host-free aquarium housed 47 clams (34 D. octotentaculata, seven D. luteocrinita, five “P”. squillina, and one D. troglodytes). The host-containing aquarium housed 46 clams (33 D. octotentaculata, seven D. luteocrinita, five “P”. squillina, and one D. troglodytes). Clams were introduced in small cohorts of conspecifics placed between the burrow openings and observed for 15 min. Those that did not enter the burrow within 15 min (i.e., remained on the sand surface or climbed the aquarium glass) were manually transferred into the burrow after this observation period. A mix of cultured unicellular green and brown algae was added to the tanks once a day and each tank was observed for patterns of spatial use, grouping behavior, and mortality. To simulate the assumed light conditions of natural stomatopod burrows, the exposed side of the artificial burrow was covered with thick black plastic bags level with the burrow entrances when observations were not taking place.

Results

Commensal occurrence and distribution

A total of 86 host burrows were sampled and 29 of these (33.7%) yielded ≥1 commensal vasconielline clam(s), collectively totaling 112 specimens from 6/7 of the known IRL vasconielline species (Goto, Harrison & Ó Foighil, 2018) and ranging in frequency from 1–21 commensals per burrow (Table S1). The burrow-wall commensal Divariscintilla octotentaculata was numerically dominant with 73 individuals sampled from 18 burrows (Fig. 6) distributed among 4/5 sampling sites (Table 1). Its ectocommensal sister species, “Parabornia” squillina, was the next most numerous with 20 individuals recovered from eight host burrows (Fig. 6), also distributed among four sampling sites (Table 1). Respective numbers for D. luteocrinita and D. troglodytes were 13 and three individuals from nine and two host burrows (Fig. 6) among three and two sites (Table 1). The products of non-adaptive within-host speciation, sister species D. yoyo and D. aff. yoyo, were the least numerous commensals recovered: respectively, two and one and both from single burrows (Fig. 6, Table 1). Sampled individuals of the four most common species ranged two-fold in mantle length (Table S2), likely representing different age classes. No specimens of the rarest IRL commensal vasconielline (Mikkelsen & Bieler, 1992), D. cordiformis, were recovered.

Figure 6 Summary frequencies of each commensal clam species recovered in the field census of Indian River Lagoon host burrows.

The total number of commensal clams recovered, and burrows occupied (in parentheses), for each of the 6 commensal species (5 Divariscintilla spp. and 1 “Parabornia” sp.) sampled from 86 IRL host Lysiosquilla scabricauda burrows.

Table 1 Summary table showing commensal species recovery from each of the five Indian River Lagoon sampling sites.

For each of the five IRL sampling sites (see Fig. 2), the respective numbers of commensal clams recovered (“Count”), and of Lysiosquilla scabricauda host burrows occupied (“Burrows”), are shown for all six commensal clam species sampled in this study: Divariscintilla octotentaculata (O), “Parabornia” squillina (S), D. luteocrinita (L), D. troglodytes (T), D. yoyo (Y), and D. aff. yoyo (AY). The two rightmost columns (“All”) collectively display the combined totals of all species of commensal clams collected at each site.

	O	S	L	T	Y	CF	All	
	Count	Burrows	Count	Burrows	Count	Burrows	Count	Burrows	Count	Burrows	Count	Burrows	Count	Burrows	
Spoil Is. 17 N	18	7	6	2	4	3	0	0	0	0	0	0	28	10	
Ft. Beach	0	0	2	2	0	0	0	0	0	0	0	0	2	2	
SE Causeway	44	7	6	2	7	5	1	1	2	1	0	0	60	11	
Coon Is.	2	1	6	2	0	0	2	1	0	0	1	1	11	3	
Spoil Is. 83 N	9	3	0	0	2	1	0	0	0	0	0	0	11	3	

Of the 29 burrows with commensals, 22 (75%) were monospecific (either Divariscintilla octotentaculata, D. luteocrinita, or “Parabornia” squillina), four had 2-species assemblages, and three had 3-species assemblages (Fig. 7). Burrows with multispecies assemblages (N = 7) shared two characteristics: they were exclusively comprised of burrow-wall commensals, and all contained D. octotentaculata individuals. The ectocommensal “P”. squillina was not recovered from any host burrow that also yielded burrow-wall commensals (Divariscintilla spp.).

Figure 7 Graphical summary of Indian River Lagoon commensal clam co-occurrence.

Census data from 29 commensal-occupied host burrows (out of a total of 86 burrows sampled) grouped by their level of commensal species diversity: mono-, bi-, and tri-specific. See Table S1 for site locations of individual burrow IDs (bar graph × axes notation) yielding ≥1 commensal clam(s).

The over-dispersion tests of “P”. squillina and D. octotentaculata rejected the null hypothesis, meaning that for each species, the observed frequency distribution was significantly (P < 0.001) more clustered than that expected by chance alone. For the comparative recruitment simulation tests (Fig. 8), when the five burrow-wall commensal species were collectively treated as one group in comparison to the ectocommensal “P”. squillina, the null hypothesis of random assembly was strongly rejected (P < 0.001). “P”. squillina was never observed to co-occur with burrow-wall commensals in the field, whereas in the simulated random recruitment scenarios there were always at least four burrows where the two groups co-occurred. Similar results were found for individual ectocommensal/burrow-wall commensal simulation tests: “P”. squillina and D. octotentaculata, as well as “P”. squillina and D. luteocrinita (Fig. 8). In stark contrast, when evaluating co-occurrence among the burrow-wall commensals D. octotentaculata and D. luteocrinita (Fig. 8), the observed field value fell well within the simulated random recruitment distribution (P = 0.405), indicating these two burrow-wall commensal species likely co-recruited to IRL host burrows following a random process.

Figure 8 Simulated random recruitment expectations for co-occurrence of commensal species in Indian River Lagoon host burrows.

Comparison of the actual observed (dashed red lines) co-occurrence of four commensal vasconielline species combinations in IRL host burrows to their simulated co-occurrence distributions (histograms) expected under random larval recruitment and post-larval survival dynamics.

Stable isotope analyses of dietary niche

Individuals of the six IRL commensal species sampled, together with samples of their potential basal resources (host tissue, burrow-water POM, and burrow-wall sediment), were analyzed to determine their respective carbon and nitrogen isotopic signatures (Table S3). Only three of the six commensals—Divarscintilla octotentaculata, “Parabornia” squillina and D. luteocrinita—were recovered in sufficient numbers to generate isotopic analysis Bayesian-estimated Standard Ellipse Areas (SEAB) plots, and their respective SEAc values were 0.184, 1.128, and 0.255. The corresponding SEAc values for host tissue, burrow-water POM, and burrow-wall sediment samples were respectively 0.826, 3.354, and 7.336.

Divariscintilla octotentaculata’s inferred isotopic niche space (its Bayesian-estimated ellipse area) was distinct from that of D. luteocrinita (Fig. 9), a fellow burrow-wall commensal, but it overlapped substantially with that of “Parabornia” squillina (Fig. 9), its ectocommensal IRL sister species. All three commensal species did not overlap in isotopic niche space with any of their three potential basal resources but they placed closest to burrow-water POM, and furthest away from the host Lysiosquilla scabricauda (Fig. 9).

Figure 9 Inferred isotopic niche widths of commensal species and potential basal resources.

Bayesian-estimated Standard Ellipse Areas (SEAB) of three IRL commensal clam species—Divariscintilla octotentaculata (N = 10; 1 sample failed), D. luteocrinita (N = 4) and “Parabornia” squillina (N = 20)—together with that of their potential basal resources: suspended particulate organic matter (POM, N = 33; 2 samples failed), deposited organic matter (sediment; N = 33; 1 sample failed), and mantis shrimp (Lysiosquilla scabricauda; N = 19). Three other IRL commensals (D. troglodytes, D. yoyo, and D. aff. yoyo) were not sampled in large enough quantities to produce SEAB plots for this analysis. X axis units, expressed as δ15N, are the ratios of 15N to 14N obtained from the labelled samples, whereas Y axis units, expressed as δ13C, are the corresponding ratios of 13C to 12C. See Table S3 for individual specimen isotopic data values and sampling details.

Artificial burrow observations

Individuals of all four commensal species (Divarscintilla octotentaculata, “Parabornia” squillina, D. luteocrinita and D. troglodytes) introduced into aquaria containing artificial host burrows (with and without a mantis shrimp host) preferred firmer surfaces to the loosely packed aquarium surface sand. Most clams that encountered the edge of an artificial burrow opening crawled down that burrow and most that encountered the aquarium glass wall crawled up that surface (prior to being manually relocated into the artificial host burrow).

In all three observations of the host-free aquarium (Figs. 10A–10C), commensal clam species were partially intermixed throughout the artificial burrow. The most numerous species, Divarscintilla octotentaculata, exhibited the clearest spatial aggregation with most individuals dominating the left 1/3rd of the burrow, leaving the rest of the burrow occupied primarily by a mixture of Parabornia squillina and D. luteocrinita (Figs. 10A–10C). Most individuals, regardless of species, were located within the horizontal segment of the artificial burrow, primarily attached to the lateral and upper burrow walls. Commensals were typically sedentary during observation periods, though there were changes in individual positioning between observations and two D. octotentaculata specimens left the burrow and attached to the aquarium walls.

Figure 10 Artificial burrow laboratory observations of commensal clam behavior and survival with and without a host.

A series of time-specific observations of the spatial positioning and survival of four Indian River Lagoon commensal clam species within experimental artificial burrows without (left), and with (right), a resident Lysiosquilla scabricauda mantis shrimp host. At the beginning of the experiment (Time 0), 47 clams were introduced to the host-free burrow (34 Divariscintilla octotentaculata, seven D. luteocrinita, five “Parabornia” squillina, and one D. troglodytes), and 46 clams were introduced to the host-occupied burrow (33 D. octotentaculata, seven D. luteocrinita, five “P”. squillina, and one D. troglodytes). N denotes the number of surviving clams observed for each treatment at the accompanying time point. This figure is an original artwork by John Megahan.

In the host-containing aquarium, the initial disturbance associated with commensal introduction led the host mantis shrimp to attempt to cover up the light-exposed glass with a sand-mucus mixture. Following this, it rested within the burrow and the commensal clams gradually positioned themselves around it. By the first observation period, 15 h post-introduction (Fig. 10D), most commensals had formed a mixed species assemblage in the horizontal segment of the artificial burrow where the host primarily rested (although three Divarscintilla octotentaculata individuals had exited the burrow and were attached to the aquarium walls). Most burrow-wall commensals attached to the upper burrow wall where many engaged in characteristic “yo-yo” behavior in response to being touched by the host. Within-burrow positioning of the five ectocommensal “Parabornia” squillina individuals varied from observation to observation (Figs. 10D–10F). Although none immediately moved onto the host, at +15 h (Fig. 9D) three had attached to the base of the host pleopods and two were attached to the upper burrow wall. At +27 h (Fig. 10E) two P. squillina individuals remained attached to the host and three to the burrow wall (two upper, one lower), and at +63 h (Fig. 10F) one individual remained attached to the host.

Commensal survivorship in the experimental artificial burrows decreased with time and two quantitatively and qualitatively distinct patterns of commensal mortality were evident during the observation period (Figs. 10A–10F). One pattern was independent of host presence: in 5/6 burrow observations (Figs. 10A–10E), a “background” rate of mortality, ranging from 1-8 individuals per time increment, was characterized by the presence of dead clam bodies lying on the bottom of the burrows. In contrast, a greatly elevated mortality rate (34/35 commensals) was detected in the +63 h observation of the host-occupied burrow (Fig. 10F), characterized by the absence of observable dead clam bodies or clam tissue/shell fragments within or outside the burrow. The sole survivor was a single individual of the ectocommensal “Parabornia” squillina attached to the base of the host pleopods (Fig. 10F).

Discussion

Our study investigated the regulation of IRL Lysiosquilla scabricauda commensal species coexistence using three complementary approaches and it uncovered a complex mix of congruence and incongruence with both neutral model (Hubbel, 2001) and competitive exclusion principle (Grinnell, 1904; Hardin, 1960; Chesson, 2003) expectations.

Some of this complexity was evident in the field census results for individual IRL host burrows for which competitive exclusion principle expectations are of co-occurrence of commensals occupying distinct host niches (burrow-wall commensals and ectocommensals) and neutral theory expectations are of a mix of commensals with and without distinct host niches. Mixed-niche commensal assemblages (predicted by both models) were absent (P < 0.001), and all observed cases of multi-species co-occurrence were exclusively composed of burrow-wall commensals, two of whom met random co-recruitment expectations (Figs. 7 and 8). We were particularly interested in the coexistence dynamics of two constituent sister species pairs alternatively generated by adaptive and by non-adaptive speciation. A reciprocal, robustly negative recruitment effect was apparent for the adaptive sister species pair: burrow-wall commensal Divariscintilla octotentaculata on ectocommensal Parabornia squillina, and vice versa, (P ≤ 0.001); a result explicitly incompatible with competitive exclusion principle expectations. Unfortunately, the rarity of the non-adaptive sister species pair [Divariscintilla yoyo (N = 2) and D. aff. yoyo (N = 1)] precluded meaningful statistical analyses of their census data.

Most host burrows sampled (66%) lacked detectable commensal clams (Fig. 7) implying that host individuals and resources may not be limiting factors for commensals. However, some of the sampled burrows without commensals could also be (1) occupied but undetected because of incomplete sampling; (2) empty because aspects of commensal life history, e.g., mating behavior (Mikkelsen & Bieler, 1992), promote within-species clustering, (as implied by the over dispersion test results for Divariscintilla octotentaculata and “Parabornia” squillina); or (3) subsets of host burrows may be otherwise inhospitable for commensal species recruitment/survival. We know that incomplete sampling was an issue for both burrow-wall commensals and ectocommensals. As noted by Mikkelsen & Bieler (1989) and Mikkelsen & Bieler (1992), yabby pumps (used to sample burrow-wall commensals) are ineffective in sampling the deeper, horizontal sections of Lysiosquilla scabricauda burrows. In addition, burrows in this stomatopod genus are typically occupied by a resident male–female monogamous pair (Christy & Salmon, 1991) and our host bait-and-capture method (Fig. 3), used to sample ectocommensals, was effective only in capturing one resident host/burrow, most likely the resident male (Ahyong, Caldwell & Erdmann, 2017).

Nevertheless, it is important to note that key aspects of our field census results are consistent with Mikkelsen & Bieler’s (1992) sampling of these same populations 30 years earlier (their fieldwork occurred in 1987), despite major IRL ecological changes (involving extensive eutrophication, algal blooms and seagrass habitat loss) in the interim (Morris et al., 2022). These include the numerical dominance of Divariscintilla octotentaculata (recorded from 88% of host burrows containing burrow-wall commensals by Mikkelsen & Bieler (1992) versus 85% in our study), its co-occurrence with other burrow-wall commensals (80% versus 38%, respectively), the absence of co-occurring burrow-wall commensals and ectocommensals (0% versus 0%, respectively) and the prevalence of commensal-free host burrows (“most” versus 66%, respectively). Further comparisons of the same metric among the two studies indicate that D. leucocrinita may have increased in occurrence (22.8% of host burrows containing burrow-wall commensals recorded by Mikkelsen & Bieler (1992) versus 42.8% in our study), but that the remaining burrow-wall commensals appear to have declined: D. yoyo + D. aff. yoyo (57% versus 9.5%, respectively; note that Mikkelsen & Bieler (1992) were unaware of D. aff. yoyo’s existence), D. troglodytes (54% versus 9.5%, respectively) and D. cordiformis (5.7% versus 0%, respectively). The collective >80% decrease of the non-adaptive sister species pair D. yoyo + D. aff. yoyo implies that their relative rarity may be a recent development.

Regarding trophic niche differentiation, neutral theory allows co-existence irrespective of trophic niche overlap, whereas competitive exclusion principle expectations are that co-existing commensals will occupy distinct trophic niches. Combined stable isotope/field census data were available for only three commensal species and the results were mixed. The only multispecies combination observed among the three—co-occurring burrow-wall commensals Divarscintilla octotentaculata and D. luteocrinita (Fig. 7, 6/7 multispecies assemblages)—exhibited qualitative separation in their isotopic niches (Fig. 9), thereby conforming with competitive exclusion principle expectations. In contrast, the other possible known heterogeneous trophic niche combination—burrow-wall commensal D. luteocrinita and ectocommensal Parabornia squillina (Fig. 9)—was not detected in any IRL host burrow (Fig. 7).

A consumer’s stable isotope composition is shaped by that of the species it consumes in a broadly predictable manner: empirical studies have shown that in consumer tissues, the ratio of 15N to 14N is generally 2.5–5 greater, and the ratio of 13C to 12C is generally similar or as much as one greater, than that of their diets (Bearhop et al., 2004). Applying this general expectation to our commensal species stable isotope data (Fig. 9) yields a pronounced mismatch with host tissue SEAB, implying that host tissues/wastes/food scraps are not significant trophic resources for the three commensals, including the ectocommensal “Parabornia” squillina. Of the two remaining putative commensal trophic resources tested (burrow-wall deposited organic material and burrow-water POM) the placement of burrow-water POM SEAB (Fig. 9) is most consistent with it being the commensal’s primary trophic resource, a conclusion in agreement with Mikkelsen & Bieler’s (1989) description of the IRL commensal species as “filter-feeders”. The qualitative isotopic niche separation shown by Divariscintilla luteocrinita from the other two commensal species (Fig. 9) could stem from a variety of factors including qualitative differences in (1) the subset of burrow-water POM material being assimilated; (2) how their respective microbiomes process ingested material; (3) accession of another basal resource (e.g., dissolved organic matter) that was not sampled in our study.

Our artificial burrow behavioral experiment was designed to test if a competitive exclusion principle expectation—the evolution of symbiont specialization for discrete within-host niches (Cook et al., 2002; Joy & Crespi, 2007; Althoff, 2014)—applied to other members of the IRL Lysiosquilla scabricauda commensal community in addition to the adaptive sister species pair of the burrow-wall commensal Divariscintilla octotentaculata and the ectocommensal “Parabornia” squillina (Goto, Harrison & Ó Foighil, 2018). Lysiosquilla scabricauda burrows are large enough to potentially facilitate fine-scale spatial partitioning among co-occurring burrow-wall commensals that might be undetectable by yabby pump sampling. Our results (Fig. 10) yielded no evidence of spatial segregation among burrow-wall commensals in the presence of a resident host: all three species clustered around the host’s primary resting location. However, this experiment did yield two unexpected new behavioral insights, although we cannot rule out the possibility that they are both experiment-induced artifacts.

One surprise concerned a hitherto unknown behavioral flexibility of the ectocommensal “Parabornia” squillina. In the control artificial burrow, lacking a host, 5/5 individuals attached to the burrow wall (Figs. 10A–10C). In the treatment artificial burrow, containing a host, only 3/5 assumed the ectocommensal condition and at least one of these subsequently moved off the host and attached to the burrow wall during the observation period (Figs. 10D–10F). This was somewhat surprising because Goto, Harrison & Ó Foighil (2018); Supplementary Movie S2) found that individuals detached from hosts rapidly reattach and, to our knowledge, extensive yabby pump sampling of IRL Lysiosquilla scabricauda burrows (Mikkelsen & Bieler, 1989; Mikkelsen & Bieler, 1992; Goto, Harrison & Ó Foighil, 2018; this study) have not recovered non-host-attached “P”. squillina individuals. It remains to be determined to what degree “P”. squillina clams alternate between ectocommensal and burrow-wall attachments in the wild.

The most surprising result of the artificial burrow behavioral experiment was the sudden breakdown of the commensal relationship resulting in a mass mortality of 34/35 commensals, apparently due to targeted predation by the host (Figs. 10E & 10F). The only survivor was a host-attached specimen of “Parabornia” squillina and all others, including at least 3 non-host attached “P”. squillina and an aquarium wall-attached specimen of Divariscintilla octotentaculata (Fig. 10F), were apparently consumed by the host. We cannot of course rule out the possibility that this sudden switch in host behavior was an artifact triggered by stressful artificial culture conditions (including the presence of much higher densities of commensals, and of different commensal species combinations, than we recovered from individual field burrows) and it is unclear if Lysiosquilla scabricauda also targets non-host attached commensals in the wild, and if so, under what conditions?

Galeommatoidea is a highly speciose superfamily (Bouchet et al., 2002; Paulay, 2003) and the vast majority of commensal members occur in soft-bottom habitats in association with larger, bioturbating macroinvertebrate hosts (collectively from diverse phyla) that provide a within-sediment depth refuge from predation (Li, Ó Foighil & Middlefart, 2012). Within this group, Lysiosquilla scabricauda’s commensals are exceptional in four aspects of their evolutionary ecology: species richness, predominant evolutionary origin mechanism, host trophic ecology, and potential for host predation. We currently know of eight host-specific commensals (Simone, 2001; Goto, Harrison & Ó Foighil, 2018), more than any other single galeommatoidean host to date, and this is likely an underestimate because only a tiny sliver of the host’s range—the IRL—has been studied in detail. Goto, Harrison & Ó Foighil’s (2018) phylogeny of 6/7 IRL commensals was consistent with a 5:1 ratio of within-host to host-switching speciation events, a much higher ratio than that documented in other galeommatoidean clades (Goto et al., 2012; Li, Ó Foighil & Strong, 2016). Mantis shrimp are one of the few predators to host galeommatoidean commensals (Yamamoto & Habe, 1961; Morton, 1980; Goto et al., 2012) and are the only known galeommatoidean hosts that engage in active, visual predation (Cronin et al., 2022). To our knowledge, L. scabricauda’s mass killing of 34/35 commensals, albeit in captivity (Figs. 10E & 10F), is the first report of galeommatoidean commensals being actively preyed upon by their host.

Synthesis

Collective consideration of the IRL field census, stable isotope, and captive behavioral data yields a heterogenous vista of symbiont coexistence and of symbiont exclusion. It may therefore be useful to view this study system as being composed of two distinct, superimposed patterns of commensal distribution: (1) all burrow-wall commensal species; (2) the ectocommensal species.

In this framing, the six burrow-wall commensals broadly adhere to neutral theory (Hubbel, 2001) expectations of species assembly in that they co-occur seamlessly in space and time, at least in the sections of IRL host burrows reached by yabby pump sampling (Mikkelsen & Bieler, 1992; this study). This is consistent with numerous studies that have found little evidence for competitive exclusion in marine benthic communities (Stanley, 2008; Shinen & Navarrete, 2014; Klompmaker & Finnegan, 2018). Such studies often emphasize the role of high rates of marine predation and disturbance in minimizing competition (Klompmaker & Finnegan, 2018) but another, possibly more apt, model for IRL burrow-wall commensal coexistence might be Laird & Schamp’s (2006) finding that coexistence of ≥3 competitors is possible if the competition is non-hierarchical. That important detail remains to be determined but at least one burrow-wall commensal (Divariscintilla luteocrinita) showed evidence of trophic differentiation (Fig. 9), and 4/6 appear to have fluctuated in relative frequency between 1987 and 2017 (Mikkelsen & Bieler, 1992; this study). Goto, Harrison & Ó Foighil’s (2018) vasconielline phylogeny established that the burrow-wall niche and its associated “hanging-foot” morphology is plesiomorphic among IRL commensals, implying that within-burrow coexistence may also be the ancestral condition. If so, it has proven to be remarkably stable and has survived the repeated addition of new burrow-wall commensals, mainly through within-host (ostensibly non-adaptive) speciation, but also through host switching (Goto, Harrison & Ó Foighil, 2018).

In contrast, “Parabornia” squillina’s apparent inability to coexist with other Lysiosquilla scabricauda commensals in IRL host burrows, despite its unique ectocommensal niche, is incongruent with both competitive exclusion principle and neutral theory expectations. Goto, Harrison & Ó Foighil’s (2018) vasconielline phylogeny shows that the ectocommensal niche is (1) a derived condition among IRL commensals; (2) a product of within-host adaptive speciation. In this case, within-host adaptive speciation involving clear ecological character displacement (Goto, Harrison & Ó Foighil, 2018) has apparently led to the introduction of strict ecological exclusion (and a truncation of realized niches) to a commensal community hitherto characterized by comprehensive co-existence: an inverse outcome of theoretical expectations.

The ecological factors regulating the observed IRL burrow-wall commensal/ectocommensal exclusion are currently obscure but potentially include differential recruitment to individual IRL host burrows and/or differential survival in “mixed-niche” burrow assemblages. Our field census data unfortunately could not distinguish among those possibilities because they did not include newly recruited juvenile commensals: based on prodissoconch sizes, they metamorphose out of the plankton at 350–390 µm in length (Mikkelsen & Bieler, 1989; Mikkelsen & Bieler, 1992; Fig. S1) and our smallest recovered specimen was 2.5 mm in mantle length. Resampling IRL host burrows during commensal recruitment peak periods using a sufficiently fine mesh sieve could address this deficiency. Replication of the adult exclusion pattern by juveniles, or detection of juvenile-specific “mixed-niche” burrow assemblages, would respectively support differential recruitment, or differential survival, exclusion mechanisms.

Current evidence for either potential exclusion mechanism is fragmentary at best. Regarding differential recruitment, the challenge is to explain the lack of “Parabornia” squillina recruitment to host burrows supporting multi-species burrow-wall commensal assemblages, and/or vice versa. Commensal galeommatoideans typically display positive chemotaxes to their respective hosts (Morton, 1962; Gage, 1968; Gage, 1979; Ockelmann & Muus, 1978), as apparently does “P”. squillina (Goto, Harrison & Ó Foighil, 2018). Preventing “mixed-niche” IRL recruitment of the seven IRL commensal species to the same exclusive host might require counteracting among-commensal negative chemotaxes/behaviors (burrow wall commensal species vs ectocommensal species or vice versa). Our laboratory behavior experiments (Fig. 10), show no clear evidence for such.

A variety of potential drivers, competitive and/or predatory, might contribute to differential burrow-wall commensal vs ectocommensal survival in “mixed” burrow assemblages. Note that the formal concept of competitive exclusion (Hardin, 1960) turns out to be inapplicable to this study system because it requires the species that cannot coexist—burrow-wall commensals and the ectocommensal (not subsets of the burrow-wall commensals as initially hypothesized)—to have identical niches, and this is clearly not the case (Fig. 1). As discussed above, evidence that a trophic competition driver is influential in regulating this system is mixed at best for the three commensal species with characterized trophic niches, with the sole member of the three to show trophic differentiation, Divariscintilla leuteocrinita (Fig. 9), co-occurring only with other burrow-wall commensals (Fig. 7).

A context-specific change in Lysiosquilla scabricauda predatory behavior is another potential driver of differential commensal survival i.e., selective host exclusion. Our artificial burrow behavioral experiment was unexpectedly terminated by the host-induced mass mortality of all non-host-attached commensals (Figs. 10E & 10F). That host behavioral change might have been triggered by starvation because captive mantis shrimp refused to feed on offered prey fishes (a response readily seen in the field). However, a host-starvation trigger does not explain the absence of ectocommensals in IRL burrows containing burrow-wall commensals (Fig. 7). An alternative trigger might be that the act of ectocommensal attachment itself induces a change in host predatory behavior leading to the eradication of co-occurring burrow wall commensals. This may seem far-fetched, but it is fully congruent with the observed field distribution data (Fig. 7) and mantis shrimp are behaviorally complex organisms with extraordinary visual systems (Thoen et al., 2014; Thoen et al., 2017; Franklin et al., 2017; Patel & Cronin, 2020). It could also be tested experimentally by tracking burrow-wall commensal survival in host burrows with (treatment) and without (control) added ectocommensals.

Conclusions

This study aimed to investigate how Lysiosquilla scabricauda’s extraordinary IRL galeommatoidean commensal community (Mikkelsen & Bieler, 1989; Mikkelsen & Bieler, 1992; Goto, Harrison & Ó Foighil, 2018), incorporating sympatric sister species pairs generated by adaptive and by non-adaptive speciation processes, is regulated. Although the unexpected rarity of the non-adaptive species pair did not allow us to fully address this goal, our results confirmed the presence of a trenchant ecological exclusion in this commensal community that violates both competitive exclusion principle and neutral theory expectations. This intriguing ecological puzzle is potentially resolvable through additional field sampling of commensal recruitment and additional host-commensal behavioral experiments. However, a fuller understanding of this commensal communities’ evolutionary ecology will also require its study outside of the narrow confines of the IRL. It would be particularly interesting to investigate southern Brazilian L. scabricauda commensal populations to establish if its ectocommensal, “Parabornia” palliopillata, also exhibits an ecological exclusion from regional burrow-wall commensals.

Supplemental Information

Supplemental Information 1 Scanning electron micrograph taken of the hinge area of a “Parabornia” squillina shell showing a dorsal view of the retained larval shell (Prodissoconch)

PI indicates the D-shaped Prodissoconch I (126 µm in length); PII shows the much larger Prodissoconch II (370 µm in length) characterized by commarginal growth lines; D indicates the post-larval, juvenile Dissoconch shell formed after metamorphosis, and the two *s locate the metamorphic line separating the larval and juvenile shell portions. The PI/PII length ratio of this specimen was 126/370 µm, or 34% [indicating planktotrophic larval development Ockelmann, 1965] and it falls between the respective values Mikkelsen & Bieler (1992) recorded for Divariscintilla octotentaculata (115/360 µm or 32%) and for D. luteocrinita (145/390 µm or 37%), two closely related species with confirmed planktotrophic larval development. “P.” squillina’s larval shell morphology is therefore entirely consistent with it undergoing planktotrophic larval development and dispersal.

Figure S2 Detail of modified aquarium viewing tank

Lateral view of an aquarium tank showing the white PVC support structures holding up a thin wedge of sediment that housed an artificial host burrow.

Table S1 Breakdown of IRL commensal clam recovery by individual host burrow

“Site” refers to the respective 5 sampling sites (Figure 2): S1 (Spoil Island 17N), S2 (Fort Beach), S3 (SE Causeway), S4 (Coon Island) and S5 (Spoil Island 83 N). “ID” refers to individual sampled host burrows at that study site yielding ≥1 commensal clam(s). The following abbreviations are used for the 7 known IRL vasonielline commensal species: Divariscintilla octotentaculata (Octo), D. troglodytes (Trog), “Parabornia” squillina (Squill), D. luteocrinita (Luteo), D. cordiformis (Cord), D. yoyo (Yoyo), D. aff. yoyo (AY).

Table S2 Minimum, maximum, and mean mantle lengths for each IRL vasconielline commensal species sampled

Mantle lengths are measured in millimeters and the 6 commensal species are indicated as follows: Divariscintilla octotentaculata (O), “Parabornia” squillina (S), D. luteocrinita (L), D. troglodytes (T), D. yoyo (Y), D. aff. yoyo (AY).

Table S3 Raw data for each stable isotope sample analyzed in this study

The “Burrow ID” column refers to the respective individual host burrows that samples were retrieved from, and sampling sites are identified by the respective Burrow ID pre-fixes: S1 (Spoil Island 17N), S2 (Fort Beach), S3 (SE Causeway), S4 (Coon Island) and S5 (Spoil Island 83 N) –see Figure 2. The “Type” column identifies the sample category: Sediment (burrow-wall sediment sample), Clam (commensal clam sample), Shrimp (mantis shrimp host sample), POM (burrow-water particulate organic matter sample) and the individual vasonielline commensal clam species are identified in the “Species” column as follows: Divariscintilla octotentaculata (O), D. troglodytes (T), “Parabornia” squillina (P), D. luteocrinita (L), D. yoyo (Y).

We thank Smithsonian Marine Station at Fort Pierce personnel Sherry Reed, Michael J. Boyle, William (Woody) Lee and David R. Branson for their unstinting help in specimen collection, specimen photography, and artificial burrow design. We also thank University of Michigan personnel Jacob Allgeier and Knute Nadelhoffer for help with the stable isotope analyses, Museum of Zoology artist John Megahan for his expert assistance with the Figures, Seto Marine Biological Laboratory personnel Mariko Kawamura for her help with the S.E.M. imaging, and Paula Mikkelsen and Rüdiger Bieler for their expert review of the manuscript.

Additional Information and Declarations

Competing Interests

Author Contributions

Data Availability

Jingchun Li is an Academic Editor for PeerJ.

Teal A. Harrison conceived and designed the experiments, performed the experiments, analyzed the data, prepared figures and/or tables, authored or reviewed drafts of the article, and approved the final draft.

Ryutaro Goto conceived and designed the experiments, prepared figures and/or tables, authored or reviewed drafts of the article, and approved the final draft.

Jingchun Li analyzed the data, prepared figures and/or tables, authored or reviewed drafts of the article, and approved the final draft.

Diarmaid Ó Foighil conceived and designed the experiments, analyzed the data, prepared figures and/or tables, authored or reviewed drafts of the article, and approved the final draft.

The following information was supplied regarding data availability:

The raw field data, raw stable isotope data, scanning Electron Micrograph data, details of viewing tank setup are all available in the Supplemental Files.

A field video recording of the mantis shrimp bait-and-capture technique is available at Figshare: Ó Foighil, Diarmaid (2024). Mantis Shrimp Capture Technique.m4v. figshare. Media. https://doi.org/10.6084/m9.figshare.24847938.v1.

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
