# Peer review of "Within-host adaptive speciation of commensal yoyo clams leads to ecological exclusion, not co-existence"

_PeerJ, doi:10.7717/peerj.17753_

## Round 0.1 · original submission · Minor Revisions

We have received two reviews for this manuscript and both reviewers found the study interesting and appreciated the level of details provided.

Reviewer 1 (R. Bieler) raised a few points that deserve revisions in the methods. Namely, the reviewer asked to explain how the study system reacts to overcollecting and, also, for clarifications in the text regarding sampling procedures and sample sizes. The comment regarding Parabornia squillina should also be considered.

Reviewer 2 (P. Mikkelsen) raised an important point regarding the effects of the number (density) and composition of commensals introduced in experimental burrows. This needs to be clearly justified/explained. The reviewer also made several editorial suggestions in the annotated manuscript attached.

·

Basic reporting

This is a very interesting study, using a species-rich host/symbiont system in East Florida’s Indian River Lagoon. I am obviously keenly interested in the topic as I was an author on the original study that discovered and described this association decades ago.

The project and its reporting is well developed.

Experimental design

I very much appreciate the detailed field and laboratory work and its documentation (the artificial burrow setup is brilliant).

Validity of the findings

No comment --please see some details below.

Additional comments

I only have a few queries and suggestions:

Materials and Methods:
This is a very small geographic area for which the species assemblage has been demonstrated (and it is amazing that the species composition has remained relatively stable over the decades). We don’t really know how such a system reacts to overcollecting. Can you perhaps add a few words about the potential impact of the collecting? Was the sampling limited to a particular regional subset and other areas were left undisturbed? Or, did you space the collecting by skipping over burrows to minimize depletion? This is not a criticism – I am just trying to understand the scale of sampling compared to the total area that has Lysiosquilla burrows.

From figure captions and later parts of the manuscript (first line of “Results”) it becomes clear how many burrows were sampled. I suggest mentioning these core data up front in the M&M section rather than just summarizing the collecting as "extensive".

Time of prior study:
You refer (in line 488 and elsewhere) that this study was undertaken about 25 years after the initial one. It actually is a span of 30 years – our 1990 and 1992 papers are based near-exclusively on 1987 field collections (see the material listings in the articles).

Genus-group names:
If I had not read the Goto et al. (2018) paper ahead of time, I would have been thoroughly confused by the notion in the text and figure 1 that Parabornia squillina is the sister taxon to ONE of the members of the Divariscintilla clade. Looking at Figure 1, it becomes clear that Divariscintilla as presented is not monophyletic. Either P. squillina is also a Divariscintilla species sensu lato, or Divariscintilla needs to be split up. This paper clearly is not the place for a formal solution, but the issue needs to be addressed up front. As possible solution, after such an explanation, would be to use "Parabornia" (in quotes) squillina in this paper.

Voucher material:
These species, with their unusual habitat and fragile shells, are extremely rare in accessible collections. Please voucher the remaining material in a formal collection and cite the catalog numbers in this manuscript. This aids in future reinvestigations and will make some new field collections (e.g., for certain shell-based studies) unnecessary.

Other associated fauna:
Although clearly not within the direct focus of the paper, it would be interesting to learn if your collecting also found the same two gastropod species

https://www.molluscabase.org/aphia.php?p=taxdetails&id=419653
https://www.molluscabase.org/aphia.php?p=taxdetails&id=419656

that we reported at the time from the same burrows (see https://www.biodiversitylibrary.org/page/8276821). Especially the larger of the two would have been very obvious in the sampling sieve.

Typo: Please note that commensal is misspelled as “comensal” in Figure 7

·

Basic reporting

This is a very interesting study, especially for me following on my work in the late 1980s and early 1990s. The finding of healthy burrows still occurring in Ft. Pierce Inlet is reassuring. The changes in densities of the various species are interesting – D. octotentaculata still most numerous, D. cordiformis still rare, but D. yoyo now also relatively rare (it was D. yoyo that was first brought to me by Ray Manning at SIFP). I do recall, though I do not have data to support, seeing specimen(s) resembling the morphology of D. cf. yoyo. A very enjoyable read that definitely deserves publication.

The paper is well-written, with good structure and figures.

Experimental design

The questions asked by the study are pertinent, the collecting and laboratory methods very well done – entirely impressive.

I have no experience with isotope analyses, so I cannot comment upon those methods or results.

Validity of the findings

The results are well justified based on the data.

I have no experience with isotope analyses, so I cannot comment upon those methods or results.

Additional comments

I have made numerous editorial suggestions in the manuscript, primarily copy editing (which I do professionally, so this is my habit). I really have only one substantive criticism.

The experimental burrow is brilliant and revealed much about the behavior of the clams and the mantis shrimp. As we always suspected, most of the clams are in the horizontal part of the burrow, thus not well sampled by the yabby. It did not surprise me that the host “rebelled” in the end under what must have been stressful conditions despite your best efforts (as you also mentioned). I wonder, did the shrimp ever leave the burrow and try to dig another one? My criticism/question pertains to the number and composition of commensals that you introduced to the experimental burrow. That composition and density is not one that you found in the field – could not that have been an additional stress factor, on them and on the mantis shrimp (who ultimately rebelled)? You never found Parabornia in the same burrow with Divariscintilla, and you never found more than three Divariscintilla in one burrow. Also, your burrow was (I think) much shorter than field burrows can be (by my recollection), which suggests that the density was amplified.

---

## Round 0.2 · accepted · Accept

I am pleased with the revisions made to the manuscript.